# Urinary Dialkylphosphate Metabolite Levels in US Adults—National Health and Nutrition Examination Survey 1999–2008

**DOI:** 10.3390/ijerph16234605

**Published:** 2019-11-20

**Authors:** Christina Gillezeau, Naomi Alpert, Priyanka Joshi, Emanuela Taioli

**Affiliations:** Institute for Translational Epidemiology, Icahn School of Medicine at Mount Sinai, New York, NY 10029, USA; christina.gillezeau@mountsinai.org (C.G.); Naomi.alpert@mountsinai.org (N.A.); priyanka.joshi@icahn.mssm.edu (P.J.)

**Keywords:** organophosphate, pesticide, environmental exposure

## Abstract

*Background:* Urinary dialkylphosphate metabolites are considered to be a proxy of the cumulative exposure to organophosphorus pesticides. We analyzed the urinary levels of six dialkylphosphate (DAP) metabolites in US adults, to assess the factors associated with levels of urinary metabolites, and observe the time trends. *Methods:* We analyzed the combined urinary levels of Dimethylphosphate (DMP), Diethylphosphate (DEP), Dimethylthiophosphate (DMTP), Diethylthiophosphate (DETP), Dimethyldithiophosphate (DMDTP), and Diethyldithiophosphate (DEDTP) in the National Health and Nutrition Examination Survey (NHANES) from 1999 and 2008. *Results:* Increased age and female gender were positively associated with combined levels of urinary DAP metabolites; BMI < 18.5 kg/m^2^, BMI > 25 kg/m^2^, current smoking, and later survey year were inversely associated with combined levels of DAP metabolites. Among those with at least one detectable DAP in their urine, the mean levels decreased starting in 2001, but stayed relatively stable through 2008. Although the maximum combined urinary DAP level was highest in 1999–2000, throughout all years, we observed extremely high levels of exposure for subgroups of individuals. *Conclusion:* Despite the fact that organophosphorus pesticides were banned for residential use in the US in 2006, there are still opportunities for exposure in the general population. The average urinary DAP levels have decreased over time; however, the decline appears to have plateaued in recent years, and there remains highly exposed individuals.

## 1. Introduction

Dialkylphosphate (DAP) metabolites can be measured in urine and are considered to be a proxy of the cumulative exposure to organophosphorus (OP) pesticides, as 2/3 of those registered by the United States (US) Environmental Protection Agency (EPA) are metabolized to produce one or more of the following six DAP compounds [1]—Dimethylphosphate (DMP), Diethylphosphate (DEP), Dimethylthiophosphate (DMTP), Diethylthiophosphate (DETP), Dimethyldithiophosphate (DMDTP), and Diethyldithiophosphate (DEDTP).

The EPA banned residential use of organophosphorus pesticides in 2006 [2], but their household use began declining a year or two earlier, when the cancellation plans started to take shape [3,4,5]. OPs are still used for insect control on many food crops, and as of 2013, thirty-six types were registered for use in the US. Although banned in the US, permanent tolerances of 0.1 to 50.0 ppms were established by the US EPA for residues of phosalone and OP in almonds, almond hulls, grapes, and stone and pome fruits imported from abroad [6]. There is debate regarding the effects of long term low dose OP exposure, as many OPs readily undergo conversion in the environment and the body, mainly in the liver, ultimately yielding alkyl phosphates, which are generally considered to be of relatively low toxicity [7]. However there is evidence that long-term low-dose exposure to some of these metabolites are associated with lower scores on neurobehavioral tests, attention deficit disorders, lower IQ levels, and possibly autism [8,9,10,11]. A recent study found a correlation between measured levels of OP exposure in house dust and urine samples, and decreases in neurobehavioral test scores [8]. Furthermore, studies have shown that acute OP poisoning is associated with nausea, headache, dizziness, hypersecretion, muscle twitching, and weakness as well as a decline in neurophysiological functions including impairments in abstraction, flexibility of thinking, simple motor skills, and sustained visual attention [12,13]. Currently, there are few studies examining the levels of OP exposure in a representative sample of the general population in the United States, making the magnitude of the issue difficult to determine.

The urinary measurements of DAP metabolites in a nationally representative sample of the US general population aged 20–59 years were published using the data collected from 1988–1994 in the National Health and Nutrition Examination Survey (NHANES). The median urinary concentrations significantly decreased over time, as did the frequency of detection of these metabolites [14,15].

A subsequent analysis of the NHANES data collected from 2003 to 2008 in both children and adults assessed the behavioral and personal factors associated with high urinary levels of four of the six available DAP compounds, and observed that urinary values were higher in females, in non-smokers, and in non-Hispanic whites [16].

In order to have a comprehensive view of the exposure levels in US adults, we analyzed here the urinary levels of six DAP metabolites of OP pesticides in adults from 1999 to 2008, to assess the factors associated with the combined levels of urinary DAP metabolites, and observe the trends over time.

## 2. Materials and Methods

### 2.1. Data/Study Population

NHANES is a nationally representative series of surveys administered by the Centers for Disease Control and Prevention (CDC) that combines interviews and physical examinations, and is designed to assess the health status of adults and children in the US [17]. NHANES is released in biennial cycles and includes questionnaire data on demographics, socioeconomic status, health behaviors, and health, as well as medical and dental examinations, and physiological and laboratory measurements. This study used data collected between 1999 and 2008, as these were the most recent years with continuous data for all of the DAP metabolite levels examined. All participants aged at least 18 years at the time of their survey, with measures for all six urinary DAP metabolites, were included in the sample.

### 2.2. Outcomes

The primary outcome was the creatinine-adjusted sum of the urinary levels of the six OP DAP metabolites that were measured as part of the NHANES examination—DMP, DEP, DMTP, DETP, DMDTP, and DEDTP. All the adult participants were required to fast for 9 h (if examination was in the morning), or 6 h (if examination was in the afternoon or evening) before specimen collection. The urine samples were collected from participants upon their arrival at the exam center and the urinary levels of OPs were quantified using a gas chromatography-tandem mass spectrometric method (GC-MS/MS). Methods for the measurement of urinary DAP metabolites, including quality control steps, are described in detail in the NHANES Laboratory Procedure Manual for each survey cycle and remained consistent between the survey cycles [18]. Because of varying detection limits (DL) over the years, the most conservative DL for each OP was used. The DL for DMP was 0.58, for DEP was 0.37, for DMTP was 0.55, for DETP was 0.56, for DMDTP was 0.51, and for DEDTP was 0.39 µg/L. For urinary DAP values below the DL, a value of the DL/√2 was imputed, consistent with NHANES reporting practices. The total level of DAP metabolites for each subject was calculated by summing the creatinine adjusted values of DMP, DEP, DMTP, DETP, DMDTP, and DEDTP.

### 2.3. Covariates

The covariates of interest included the 2-year survey cycle, age, gender, race/ethnicity, level of education, family poverty income ratio (PIR), occupation, BMI (kg/m^2^), current smoking status, whether the participant reported using pesticides in the home within the last week, and place of birth (US or outside US).

Age and family PIR were recorded as continuous variables. Race/ethnicity was defined as Non-Hispanic White (NHW), Non-Hispanic Black (NHB), Hispanic, and Other (including Asian and multi-racial participants). BMI was grouped according to the cutoff levels utilized by the CDC — < 18.5, 18.5–24.9, 25–29.9, and ≥ 30 kg/m^2^ [19]. The participants were classified as smokers if they reported currently smoking some days or every day; otherwise, they were classified as non-smokers. Occupation was categorized as follows—office jobs; service, labor, and construction jobs; agricultural jobs (including those in farming, forestry, and fishing); and unemployed (including individuals who did not have an occupation listed and whose reported reason for not working within the last week was—taking care of their homes or families, going to school, retirement, inability to work for health reasons, disability, or having been laid off).

### 2.4. Statistical Analysis

Univariate and multivariable linear regressions were used to assess the association between the covariates and the total level of DAP metabolites. All the variables with at least a marginally significant association with the total DAP level (*p* < 0.15) were considered for the multivariable model. Multivariable models were conducted on the subset of individuals with complete data for the included covariates. A sensitivity analysis was also conducted excluding those individuals with outlier values of total metabolite levels (>2 standard deviations above the mean). Descriptive statistics of continuous creatinine-adjusted OP measures were obtained for those who had at least one DAP metabolite level above the DL. Descriptive analyses were also conducted to examine the distribution of each metabolite separately over time, among those with values above the DL.

Statistical analysis was performed using the SAS software, version 9.4 (SAS Institute, Cary, NC, USA). In order to account for the complex sampling strategy of NHANES, the survey procedures were used for all modeling and descriptive statistics. Sample weights were assigned based on the NHANES subsample with urinary OP measures, and reweighted across survey years to be representative of the population at the midpoint of our study time frame. Boxplots were made in RStudio version 1.1.456 using the boxplot function in ggplot2.

## 3. Results

There were 7322 individuals interviewed between 1999 and 2008 who met the selection criteria. Of those, 6378 had at least one DAP metabolite above the DL and were included in the descriptive analyses examining the distribution of total metabolites over time.

The mean (SE) total DAP level for all of the subjects was 19.26 (1.01) µg/g of creatinine, although the median (IQR) was much lower at 6.81 (2.74–18.18) µg/g of creatinine. In unadjusted analyses, increased age was significantly associated with higher total DAP levels (*p* = 0.04), as was female gender (*p* < 0.001), being currently unemployed or having an office job (*p* < 0.001), a BMI between 18.5 and 24.9 kg/m^2^ (*p* = 0.02), being a non-smoker (*p* < 0.001), and being born in the US (*p* = 0.04). There was a marginally significant association between the total DAP levels and survey cycle, family PIR, and race/ethnicity; therefore these variables were also considered for inclusion in the multivariable model. There was no significant association between the total DAP levels and education or self-reported pesticide use (Table 1).

Multivariable models including all significant and marginally significant (*p* < 0.15) variables were run. Occupation and race were excluded from the final multivariable models as their adjusted *p* values exceeded the threshold for marginal significance. The final multivariable model for the mean total DAP level was adjusted for age, gender, family PIR, BMI, smoking status, place of birth, and 2-year survey cycle (*n* = 5660). Increased age remained significantly associated with a higher total DAP level (β_adj_: 0.2, 95% CI: 0.05, 0.35), as did female gender (β_adj_: 7.66, 95% CI: 3.10, 12.22). Compared to those with a BMI between 18.5–24.9 kg/m^2^, those with a BMI < 18.5 kg/m^2^ had significantly decreased total DAP levels (β_adj_: −9.53, 95% CI: −17.62, −1.44). Those with a BMI between 25–29.9 and ≥ 30 kg/m^2^ also tended to have lower total DAP levels, on average, than those with a BMI between 18.5–24.9 kg/m^2^, although these differences were not statistically significant. Smokers also had significantly lower total DAP levels (β_adj_: −7.98, 95% CI: −11.83, −4.12), compared to non-smokers. Compared to 1999–2000, the later years were significantly associated with a lower predicted total DAP. Higher family PIR trended toward lower average total DAP levels, while being born abroad trended toward higher average total DAP levels; however, neither predictor was statistically significant in the multivariable model (Table 2). In the sensitivity analysis excluding outlier values of total DAP levels, the results were similar, except that the relationship between BMI ≥25 kg/m^2^ and lower DAP levels became significant in addition to the significant relationship between BMI <18.5 kg/m^2^ and lower DAP levels (data not shown).

Among those with at least one DAP above the DL (*n* = 6378), the overall range of total DAP levels was 1.47–2374.25 µg/g of creatinine. The median total DAP level was 8.90 µg/g of creatinine (IQR: 4.0–21.3) and the mean was 22.19 µg/g of creatinine. When looking at the total DAP levels according to the sampling cycle, 90.8% of the subjects had at least one DAP metabolite above the DL in 1999–2000, while the percentage was 86.2% in 2007–2008. The maximum values of total DAP metabolites were highest in 1999–2000 (2374.3 µg/g creatinine) and lowest in 2003–2004 (318.4 µg/g creatinine). The median total DAP level was also highest in 1999–2000 (11.5 µg/g creatinine). In subsequent cycles, the median DAP levels were 6.8, 9.3, 8.1, and 9.8 µg/g creatinine for 2001–2002, 2003–2004, 2005–2006, and 2007–2008, respectively (Figure 1).

When examining each DAP metabolite separately, we found that while the percentage of people who have detectable levels of DAP metabolites decreased over time for most metabolites, the detectable levels measured in these people appeared to increase over time. For example, although the percentage of people with DMP levels at or above the DL decreased from 51% in the 1999–2000 cycle, to 38% in the 2007–2008 cycle, the median DMP level measured in the 1999–2000 cycle was 2.58 µg/g of creatinine and by the 2007–2008 cycle, the median rose to 10.87 µg/g of creatinine. The minimum levels also rose over time from 0.21 µg/g of creatinine in 1999–2000 to 1.69 µg/g of creatinine in 2007–2008, as did the maximum values 203.39 µg/g of creatinine in 1999–2000 and 327.15 µg/g of creatinine in 2007–2008. The remaining DAP metabolites showed similar trends (Appendix A). There was a statistically significant correlation between the creatinine-adjusted values of DMP, DEP, DETP, and DEDTP, but not between DMTP and any of the other OP metabolites, nor between DMDTP and DEP or DEDTP; however there were statistically significant correlations between DMDTP and DMP and DETP. The highest correlation was between DMP and DMDTP levels (r = 0.21) and the range of the remaining statistically significant correlations was between r = 0.03 and r = 0.15 (all *p*-values < 0.001).

## 4. Discussion

Based on the results of this analysis, being older or female, having a BMI between 18.5 and 24.9 kg/m^2^, being a non-smoker, and being sampled in an earlier sampling cycle were all factors associated with higher total DAP levels. Being born abroad, level of education, family PIR, and self-reported home pesticide use were not associated with increased DAP levels. Job occupation was not statistically significantly associated with DAP levels when added to the multivariable models. These results suggest that health behaviors, rather than occupational exposure may be the primary source of OP exposure.

The complex relationship with total urinary OP metabolites and BMI may be explained by the lipophilic nature of OP. Those with a BMI below 18.5 kg/m^2^ had lower average total DAP levels than those with a BMI between 18.5 and 14.9 kg/m^2^. However, our results seem to point to people with a BMI above 25 kg/m^2^ also having lower average total DAP levels. For those with a BMI below 18.5 kg/m^2^, which is the minimum BMI considered to be a healthy weight by the CDC, lower food intake, including lower intake of fresh fruits and vegetables may be one of the reasons for the lower levels. Research of the disease course in patients with OP poisoning found that those with a BMI higher than 29.9 kg/m^2^ required longer courses of treatment than those with a lower BMI because the OP was retained in the fat tissue for longer [20]. It is possible, then, that the lower OP metabolite urine levels seen in this study are not a reflection of the level of exposure for those with a higher BMI, but rather a reflection of the tendency for their increased amount of fat tissue to retain OP rather than excreting them in urine. It is also possible that people with a BMI over 25 kg/m^2^, which is considered the maximum healthy BMI by the CDC, may have worse overall health behaviors including decreased intake of fresh fruits and vegetables. Current smokers had lower average total DAP levels than non-smokers. This finding is consistent with a 2018 study of pregnant women in the Netherlands, which found that non-smoking was associated with higher OP metabolite concentrations [21]. This study, which also looked at the fruit intake in the mother’s first trimester, found that the women who smoked during pregnancy had lower fruit intake, which may play a role in this study as well. As an alternative possible explanation, a 2010 study in rats suggests that nicotine exposure may affect the metabolism of OP pesticides in the body, although the accuracy of this model has not been explored in humans [22].

The finding that increased age is associated with increased average total DAP levels is consistent with previous research that shows the elderly are at an increased risk of having higher OP metabolite levels and suggests that this may be related to an increased amount of time spent indoors at older ages [14]. Increases in health behaviors like higher fresh fruit and vegetable intake, as people age may also play a role in this difference.

Although our results were not statistically significant in the multivariable analysis, a higher family PIR appeared to be associated with decreased total DAP levels. We hypothesize that this may be a result of people with higher family PIR buying organic produce over regular fruits and vegetables, which tend to be more expensive. Further information regarding shopping habits and produce intake is necessary to be able to fully elucidate the nature and magnitude of the association between poverty and OP exposure. Based on the results, it appears that while the overall average total DAP level measured may be decreasing over time in the general population, there remains some individuals with extremely high total DAP levels, that may actually be increasing over time. This seems to suggest that over time there are fewer people at or above the DL, but those who are above the DL have increasingly high exposure. The source of such exposure is not known, although based on the multivariable results, it seems likely that health behaviors like consumption of fresh fruits and vegetables may play a significant role. Collection of data regarding fresh fruit and vegetable intake and other dietary habits from all participants should be considered in future iterations of NHANES as it would allow researchers to elucidate the role that diet plays in OP exposure. Currently, the information available is limited to adults at least 60 years old in the 1999–2000 survey cycle. Without that information, it is not possible to exclude hand-to-mouth contact or water and soil contamination as possible sources of exposure. However, given the lower persistence of OP in the environment [23], food contamination seems to be the most likely source. Furthermore, in this data, the EPA ban on residential use of OP in 2006 [2] did not result in a drop in DAP levels measured in the 2007–2008 survey cycle, indicating that further action may need to be taken to limit exposure.

This study has a major strength, the large sample size. This, combined with the oversampling methods that NHANES employs allows for a more representative sample of people than might be available in more locally based studies. To the authors’ knowledge, no other studies have been conducted using NHANES data and examining the levels of these DAP metabolites in adults between 1999 and 2008. Thus this study provides unique insight into the proportion of the population at risk from high levels of exposure to OP pesticides as well as some indication about the characteristics of those at risk for having a high exposure level.

The main limitation of this study is that none of these metabolites are specific to any type of OP exposure and can be formed as a byproduct of the normal human metabolic process; thus the urinary concentrations may overestimate the level of exposure to OP pesticides [24,25]. However, regardless of the source of DAP metabolites, the trend of increasing urinary DAP levels in those who have levels above the DL merits further investigation. The variable limits of detection in the NHANES dataset over time require that the most conservative DL be used for each pesticide. Finally, NHANES does not collect or report certain important variables including fresh fruit and vegetable intake in those under 60 years of age and the subjects’ area of residence (i.e., the urban or rural status) are also restricted in NHANES. As a result, it was not possible to assess whether there was a relationship between level of urbanity and pesticide exposure or to differentiate between exposures from dietary sources and those from environmental sources.

## 5. Conclusions

This study reports data collected from five NHANES survey periods. The results suggest that women, older adults, people with a BMI between 18.5 and 24.9 kg/m^2^, and non-smokers are at increased risk of having high levels of DAP metabolites in their urine, potentially indicating high levels of exposure to OP. Over time, it appears that the number of people with these high levels of exposure decreases, but the levels of exposure may increase over time in those who are highly exposed. This suggests that additional precautions may be necessary for this select group with high exposure levels. Further research to examine the sources of this high exposure is necessary to identify the specific precautions that should be taken.

## Figures and Tables

**Figure 1 ijerph-16-04605-f001:**
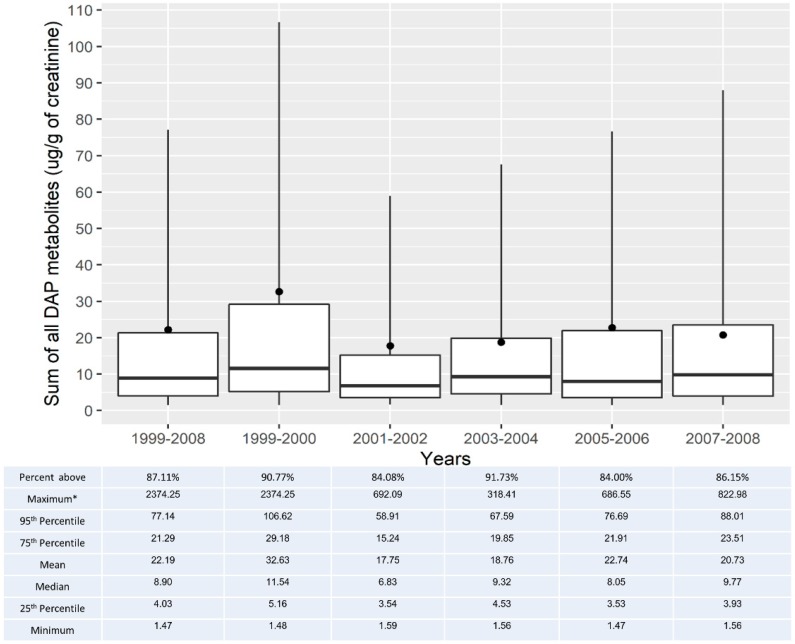
Time trends in the sum of all urinary DAP metabolites in adults from 1999 to 2008 with at least one DAP metabolite at or above the DL. Notes: Boxplots represent the minimum, first quartile, median, third quartile, and 95th percentile values. Mean values are represented by the black circle. * Indicates value not depicted in the figure.

**Table 1 ijerph-16-04605-t001:** Statistics and univariate values for adults sampled in the NHANES study between 1999 and 2008.

Variable	*n* = 7322	%	Mean DAP Value	SE	*p* Value
Age (years) Mean (SE)	42.94	0.29			0.04
0–25th percentile	18–29.91		19.72	3.3	
25th percentile- median	29.92–41.35		15.02	0.98	
median-75th percentile	41.36–52.60		18.2	1.56	
75th percentile +	52.61+		24.02	1.52	
Sex					*p* < 0.001
Female	3821	52	23.11	1.89	
Male	3501	48	15.21	0.85	
Race/Ethnicity					0.08
White	3343	46	19.85	1.4	
Hispanic	2080	28	18.02	1.57	
Non-Hispanic Black	1563	21	15.46	0.88	
Other	336	5	22.4	4.04	
Education					0.37
<High School	2289	31	17.29	1.22	
High School Grad/GED	1769	24	18.74	1.4	
Some College or AA degree	1973	27	17.62	1.39	
>College Grad	1282	18	23.43	3.22	
Missing	9	0	3.09	0.75	
Family PIR mean (SE)	3.03	0.04			0.11
0–25th percentile	<1.52		22.8	3.28	
25th percentile-median	1.52–3.07		18.34	1.27	
median-75th percentile	3.08–4.98		18.73	1.27	
75th percentile+	4.99+		17.43	1.05	
Occupation					*p* < 0.001
Office Job	2166	30	20.09	2.24	
Service, Labor, Construction	2087	29	15.15	0.8	
Farming, Forestry, Fishing (Agriculture)	96	1	13.68	2.01	
Unemployed *	2775	38	22.57	1.38	
Missing	198	3	14.98	3.31	
BMI (kg/m^2^)					0.02
<18.5	147	2	13.95	1.95	
18.5–24.9	2316	32	21.93	1.35	
25–29.9	2388	33	18.03	0.91	
>30	2372	32	19.34	2.41	
Missing	99	1	13.84	1.91	
Smoking Status					*p* < 0.001
Current Smoker	1554	21	13.41	0.76	
Non Smoker	5006	68	21.46	1.33	
Missing	762	10	14.82	1.41	
Place of Birth					0.04
US	5598	77	24.06	2.44	
Abroad	1722	24	18.39	1.08	
Missing	2	0	2.77		
Pesticide Use in Home					0.56
Yes	1091	15	18.1	1.3	
No	5906	81	19.41	1.19	
Missing	325	4	20.61	3.02	
Year					0.08
1999–2000	964	13	29.21	5.13	
2001–2002	1294	18	14.38	1.59	
2003–2004	1559	21	17.03	1.22	
2005–2006	1613	22	17.41	1.54	
2007–2008	1892	26	20.62	1.57	
Creatinine (mg/dL)	127.27	1.56			DAP values individually adjusted
0–25th percentile	<63.73		27.97	2.18	
25th percentile-median	63.73–114.14		20.02	1.38	
median-75th percentile	114.15–172.78		17.85	2.69	
75th percentile+	172.79 +		11.45	0.64	

Notes: GED-General Education Development. AA-Associates Degree. * Unemployed includes individuals who did not work within the last week because they were taking care of a house or family, going to school, retired, unable to work for health reasons, laid off, or disabled.

**Table 2 ijerph-16-04605-t002:** Adjusted model estimate in adults sampled in the NHANES study between 1999 and 2008.

Variable	Beta Estimate	Lower 95% CI	Upper 95% CI
Age (years)	0.2	0.05	0.35
Sex			
Male	REF	REF	REF
Female	7.66	3.1	12.22
Family PIR	−1.64	−3.33	0.05
BMI (kg/m^2^)			
18.5–24.9	REF	REF	REF
<18.5	−9.53	−17.62	−1.44
25–29.9	−3.87	−7.88	0.13
>30	−3.73	−10.52	3.07
Smoking Status			
Non Smoker	REF	REF	REF
Current Smoker	−7.98	−11.83	−4.12
Place of Birth			
US	REF	REF	REF
Abroad	5.07	−1.11	11.25
Year			
1999–2000	REF	REF	REF
2001–2002	−14.97	−27.13	−2.82
2003–2004	−14.34	−26.11	−2.57
2005–2006	−13.73	−25.76	−1.7
2007–2008	−11.41	−23.67	0.85

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
