# Peer review of "Urinary Dialkylphosphate Metabolite Levels in US Adults—National Health and Nutrition Examination Survey 1999–2008"

_ijerph, 2019, doi:10.3390/ijerph16234605_

Round 1

Reviewer 1 Report

The authors made conclusions on the exposure of organophosphorus pesticides on US adults. They used only NHANES data and selected specific groups of volunteers. I think that there is not enough information for a detailed statistical analysis. Authors could extend their literature review since there is much information on the levels of dialkyl phosphate metabolite worldwide. Paper is very well written, data and methods are very well documented. I could recommend this paper to be published but with a much-extended literature review.

Author Response

We would like to thank the reviewer for their comments.  Per the reviewer’s comment we have extended the literature review and attempted to clarify that this analysis is only meant to comment on OP exposure in the US. Within the US, NHANES is a nationally representative sampling of the US population, and we did not select any specific groups within NHANES, but rather limited the analysis to adults who had complete DAP levels reported, which was accounted for using the survey procedures.  Since the study only comments on OP exposure in US adults, this seems like an appropriate sample and with a sample size of over 7,000 people, there are certainly sufficient numbers to run statistical analysis. 

Reviewer 2 Report

This is an interesting study that uses the US NHANES survey to consider the exposure of US adults to organophosphate pesticides by examining the concentration of dialkylphosphate metabolites in urine samples. The article does fall into the scope of the journal and is likely to be of interest to the journals readership.

The manuscript is reasonable well written but would benefit with a close look of the language used with a view to replacing that which is colloquial with more formal or scientific terms. For example at line 91 the term ‘laid off’ is used which would be better written as ‘made redundant’ or ‘had their employment suspended or terminated’.  It would also read much better if use of the first person was removed (e.g. we, our). Also make sure all abbreviations are spelt out in full before they are first used e.g. CDC.

The rational for the study also needs to be significantly strengthened. Understanding the level of exposure does not explain the heath risks and, as the authors mention at the end of the study, some of these types of metabolites can be formed naturally in the body. Greater discussion of the possible exposure routes (food, water, spray drift, dislodgable residues, domestic use of products for pest management etc), how much of that detected might be due to natural processes and at what level does the concentration exceed natural levels and become a health risk. Additional discussion of what these health risks might be is also needed.

Information is missing on the urine sampling and analysis process. When was the sample taken? Had the patient fasted? What was the analytical process?. What measures were taken for QA?, what was the analytical LOD/LOQ. I appreciate that this was survey data but nevertheless this information an be tracked down and is vitally important for the interpretation of the data.

Author Response

We thank the reviewer for their comments.  We reviewed the manuscript again and fixed any unidentified abbreviations.  In response to the comment regarding language, we choose to utilize a first-person active voice in order to ensure clarity and avoid the confusion and additional length that sometimes results from passive voice.  As this is an established scientific writing style and IJERPH does not have a prohibition against it, we elected to err on the side of brevity and clarity. In regards to terms like “laid off” in line 114, in order to assist researchers in reproducing the results, the authors retained the language used in the NHANES survey. 

We broadened our discussion of the effects of both acute and long-term exposure to OP to help explain why establishing the levels of exposure in the average person is necessary in lines 38-50.   We also broadened our discussion regarding potential routes of exposure and the reasoning behind our conclusion that OP exposure is at least partially to blame for the rise in DAP metabolite levels in lines 245-253.  Unfortunately, there is a paucity of data regarding the levels of DAP in the average person from natural processes.  We suspect this is because it is difficult to disentangle the source of the DAP metabolites.  We added a brief explanation in lines 83-94 regarding the sampling and analysis process conducted by NHANES as well as adding the citations for the laboratory manual per the reviewer suggestion.